# Multi-Target Effects of ß-Caryophyllene and Carnosic Acid at the Crossroads of Mitochondrial Dysfunction and Neurodegeneration: From Oxidative Stress to Microglia-Mediated Neuroinflammation

**DOI:** 10.3390/antiox11061199

**Published:** 2022-06-18

**Authors:** Roberto Iorio, Giuseppe Celenza, Sabrina Petricca

**Affiliations:** Department of Biotechnological and Applied Clinical Sciences, University of L’Aquila, Via Vetoio, 67100 L’Aquila, Italy; giuseppe.celenza@univaq.it (G.C.); sabrina.petricca@univaq.it (S.P.)

**Keywords:** ß-caryophyllene, carnosic acid, phytochemicals, neuroinflammation and oxidative stress, neuroglia, neurodegeneration, CB2R/PPARγ pathway, Keap1/Nrf2/ARE transcription pathway, NLRP3 inflammasome, mitochondrial protection, PINK/parkin and mitophagy, mitochondrial dynamics and biogenesis

## Abstract

Inflammation and oxidative stress are interlinked and interdependent processes involved in many chronic diseases, including neurodegeneration, diabetes, cardiovascular diseases, and cancer. Therefore, targeting inflammatory pathways may represent a potential therapeutic strategy. Emerging evidence indicates that many phytochemicals extracted from edible plants have the potential to ameliorate the disease phenotypes. In this scenario, ß-caryophyllene (BCP), a bicyclic sesquiterpene, and carnosic acid (CA), an ortho-diphenolic diterpene, were demonstrated to exhibit anti-inflammatory, and antioxidant activities, as well as neuroprotective and mitoprotective effects in different in vitro and in vivo models. BCP essentially promotes its effects by acting as a selective agonist and allosteric modulator of cannabinoid type-2 receptor (CB2R). CA is a pro-electrophilic compound that, in response to oxidation, is converted to its electrophilic form. This can interact and activate the Keap1/Nrf2/ARE transcription pathway, triggering the synthesis of endogenous antioxidant “phase 2” enzymes. However, given the nature of its chemical structure, CA also exhibits direct antioxidant effects. BCP and CA can readily cross the BBB and accumulate in brain regions, giving rise to neuroprotective effects by preventing mitochondrial dysfunction and inhibiting activated microglia, substantially through the activation of pro-survival signalling pathways, including regulation of apoptosis and autophagy, and molecular mechanisms related to mitochondrial quality control. Findings from different in vitro/in vivo experimental models of Parkinson’s disease and Alzheimer’s disease reported the beneficial effects of both compounds, suggesting that their use in treatments may be a promising strategy in the management of neurodegenerative diseases aimed at maintaining mitochondrial homeostasis and ameliorating glia-mediated neuroinflammation.

## 1. Introduction

The combination of genetic, environmental, and sociodemographic factors, as well as unhealthy lifestyles, including a high-calorie diet and physical inactivity, cigarette smoking, alcohol consumption, recreational drug use, and the overuse and misuse of communication technology can lead to the development of chronic diseases, recently identified as non-communicable diseases (NCDs) [1,2]. Atherosclerosis, cardiovascular and chronic respiratory diseases, cancer, diabetes, and neuropathological conditions, such as Alzheimer’s disease (AD) and Parkinson’s disease (PD), account for approximately 82% of NCD-related mortality worldwide [2,3]. Due to their negative impact on the economic growth, social equity, and environmental protection, NCDs have been recognized as a huge challenge for sustainable development. Generally, these disorders are characterized by slow progression and long duration as clinical symptoms become noticeable only after substantial cellular injury has affected the target tissues. Accordingly, the development potential for NCDs significantly rises as the population ages, posing major challenges for healthcare sectors [4]. Therefore, the World Health Organization (WHO, Geneva, Switzerland) assigned a high priority level for the prevention and control of NCDs [3].

Oxidative stress (OS) and inflammatory responses (IRs) are interdependent processes, closely related to chronic diseases, including NCDs [5,6,7,8,9,10]. OS is an important phenomenon in cells and tissues reflecting an imbalance between the production of pro-oxidant species and the ability of a biological system to counteract/detoxify their harmful effects through antioxidants. This specific condition can lead to cellular damage affecting intracellular structures, including protein, lipids, and DNA, as well as mitochondrial function, extra-cellular matrix remodelling, and cell growth [9]. The exposure to chronic OS induces a subclinical state of increasing cell damage that builds up over time, facilitating tissue alteration/remodelling, selective system failures, and finally disease [7,9,11]. On the other hand, OS-induced damage to tissues can also trigger an IR and thus the release of inflammatory mediators, which in turn can directly induce OS [12,13]. Indeed, OS can trigger the activation of redox-sensitive transcription factors, such as activator protein 1 (AP-1) and nuclear factor kappa-B (NF-κB), and the subsequent recruitment of inflammatory cells to the damage site, resulting in the release of several pro-inflammatory factors, including NADPH oxidase, arachidonic acid, prostaglandins, myeloperoxidase, chemokines, and cytokines, leading to the event called the “cytokine storm” [14,15]. In turn, the increased levels of pro-inflammatory cytokines result in the hepatic generation of acute-phase proteins, including C-reactive protein (CRP). Its release and secretion further fuel the recruitment and accumulation of inflammatory cells to the damage site, accelerating the production of reactive species and, thereby, further stimulate ongoing OS [7,12,13]. Therefore, the vicious circle linking OS and IRs sustains and amplifies all stages of the disease.

Cell systems have evolved many mechanisms to lessen OS and mitigate reactive oxygen species (ROS) damage. The transcription factor Nrf2 (nuclear factor erythroid 2-related factor 2) together with its cytoplasmic repressor Keap1 (Kelch-like ECH-associated protein 1) represent the master regulators of cellular stress responses, managing redox homeostasis, cytoprotective programming, and anti-inflammatory activities. Under normal physiological conditions, Nrf2 is sequestered by Keap1 (via its cysteine thiols) in the cytoplasm, in a “silent” form, and is quickly turned over by the ubiquitin proteasome system (by recruitment of a Cul3 ubiquitin ligase) via the BTB (Broad complex, Tramtrack and Bric-à-Brac) domain of Keap1. However, when cells experience OS, or in the presence of electrophilic species (the so called “Nrf2 activators”, e.g., carnosic acid, and beta caryophillene for their relevance in this review), Nrf2 rapidly disassociates from Keap1 and translocates to the nucleus, triggering the transcription of more than 250 homeostatic and anti-oxidative genes (e.g., NAD(P)H qui-none oxidoreductase 1 (NQO1), and heme-oxygenase (HO-1)) by binding to AREs (antioxidant response elements) or EpRE (electrophile responsive element). On the other hand, ROS-generated OS, along with NF-κB activation, also acts as an upstream signal for NLRP3 (nucleotide-binding domain, leucine-rich family, pyrin domain-containing 3) inflammasome, an intracellular multiprotein complex that plays a central role in the innate immune responses and in the onset and progression of different NCDs, especially neurodegenerative diseases.

Mitochondria are key sensors of multiple types of stress [16,17,18] and important modulators of innate immunity pathways. The accumulation of defective mitochondria, due to impaired mitophagy, generation of OS, and aberrant deposition of proteins and/or misfolded protein aggregates, represent common denominators linking neuroinflammation and the pathophysiology of neuronal degeneration. Therefore, the accumulation and release of mitochondrial danger-signalling products (ROS, DNA, proteins, and lipids) are drivers of NLRP3 inflammasome activation.

Once assembled and activated, NLRP3 induces the proteolytic cleavage and release of pro-inflammatory cytokines, including interleukin (IL)-1β and IL-18, resulting in pyroptotic cell death, thereby exacerbating inflammatory responses. In this regard, the upregulation of the Keap/Nrf2/ARE signalling pathway and the orchestrated activation of HO-1 and NQO1 have protective and inhibitory effects on NF-κB activity and NLRP3 inflammasome activation. Moreover, recent findings strongly support a role of Nrf2 in the maintenance of mitochondrial function and fitness, including mitochondrial respiration, dynamics, biogenesis, and mitophagy [19,20,21,22,23], independent of transcriptional modulation of antioxidant and detoxification genes. Therefore, dysfunctions in the Nrf2/ARE signalling pathway and the aberrant activity of the NLRP3 inflammasome, along with chronic neuroinflammation and the loss of mitochondrial homeostasis, are interconnected mechanisms strictly associated with the onset and progression of neurodegenerative diseases, being their major driving force.

Although many aspects of inflammation-driven neurodegeneration remain unclear, recent research has identified glial cells, especially microglia (the brain-resident immune cells) and blood-derived monocytes, as key mediators of neurodegeneration [24]. In fact, together with NLRP3-induced enhanced expression of IL-1β by the central nervous system (CNS), microglial morphological alterations constitute the signatures associated with human neuropathology. Their plasticity in adopting multi-variated functional phenotypes decrees neuropathophysiology. In this regard, overactivated or dysfunctional microglia (M1 phenotype, pro-inflammatory) can cause deleterious effects on neuronal cells through the activation of potent neuroinflammatory reactions characterized by the excessive release of different pro-inflammatory cytokines, including IL-1β, IL6, IL-18, and tumour necrosis factor-alpha (TNF-α), and other inflammatory mediators, including reactive oxygen and nitrogen species (ROS and RNS) [25,26,27]. Microglia-driven neuroinflammation, if not resolved, alters microglia–astrocytes–neuron crosstalk and homeostasis, ultimately leading to the pathogenic evolution of neurodegeneration and development of AD and PD. In this regard, glia-regulated neuroinflammation and mitochondrial dysfunctions are interdependent, sharing converging neurodegenerative signalling. Therefore, damaged mitochondria can trigger glial inflammatory responses, and pro-inflammatory mediators can further boost negative impacts on mitochondria, causing deleterious effects on neuronal cells.

In order to break the vicious circle of neuroinflammation and minimize neuronal damage, it is essential to manage/modulate microglial immune-related functions. Deactivating the M1 phenotype and/or inducing neuroprotective microglia (M2 phenotype with anti-inflammatory activity) may ameliorate neurodegeneration.

In this regard, regulating signalling through cannabinoid receptors (CBR) is emerging as a promising strategy for switching off chronic inflammation in neurodegeneration. Over the last two decades, the endocannabinoids (eCBs) have emerged as crucial mediators of different aspects of human health and disease [28]. They are pleiotropic bioactive lipids displaying multiple and relevant neuro-immune modulatory properties [29]. In particular, the neuroprotective effects of endogenous, synthetic, and plant-derived eCBs have been robustly documented. Interestingly, microglia express all components of the endocannabinoid system (ECS) comprised of eCBs, the metabolic enzymes responsible for the synthesis and degradation of eCBs, and the cannabinoid receptors CB_1_ and CB_2_ [30,31]. Notably, upon activation, microglia adopt a neuro-protective phenotype, enhancing their synthesis of eCBs and upregulating the expression of CB_2_ receptors. Moreover, many studies have described that upregulation of eCBs signalling mitigates microglial overactivation and is effective in ameliorating neurodegeneration in different neurological disorders [32].

Therefore, identifying bioactive compounds that can target microglial CB_2_ receptors/Nrf2 signalling pathway to dampen the neuroinflammatory response [33] and/or counter mitochondrial oxidative damage, as well as enhance immunity and attenuate inflammatory modulators, may be a useful approach in preventing or delaying the onset of chronic diseases, particularly neurological disorders. In this perspective, compelling evidence has demonstrated that dietary patterns rich in plant foods are correlated with improved health and contribute to lowering the risk of NCDs, such as cardiovascular and neurodegenerative diseases, some cancers, and diabetes [34,35,36]. Therefore, due to their considerable benefits in the prevention and management of modern diseases, the use of bioactive phytochemicals, including terpenes and terpenoids, has recently gained growing interest [37,38,39,40].

In particular, bioactive ligands, such as β-caryophyllene (BCP) and carnosic acid (CA), may be promising compounds for future clinical applications due to their extremely low toxicity and multitarget mechanism of action. BCP is a dietary phytocannabinoid, with high cannabinoid type 2 receptor (CB_2_R) selectivity [41,42], shown to exert CB_2_R mediated antioxidant, immunomodulatory, and anti-inflammatory activities [43,44,45,46,47,48]. Among the multiple mechanisms triggered by BCP binding to CB_2_R, the activation of nuclear peroxisome proliferator-activated receptors (PPARs) has been clearly described. In this regard, a direct interaction of BCP with PPAR-α has also been demonstrated.

CA is a phenolic diterpene exhibiting direct and indirect antioxidant properties [49]. Notably, CA can interact and activate the Keap1/Nrf2 signalling pathway, resulting in the induction of endogenous antioxidant “phase 2” enzymes and the subsequent reduction of NLRP3 inflammasome activation, along with the suppression of microglia-mediated inflammation and improved mitochondrial homeostasis. CA and BCP also exert their protective effects by inhibiting important inflammatory mediators, such as IL-1β, IL-6, inducible nitric oxide synthase (iNOS), TNF-α, NF-κB, and cyclooxygenase 2 (COX-2).

In this review, we highlight the potential of BCP and CA in reducing OS and inflammation, with a specific focus on neuropathology, including AD, PD, and retinal degeneration. Therefore, after summarising the basic aspects of redox biology and inflammation, we provide a comprehensive overview of the molecular mechanisms, signalling pathways, pharmacological properties, bioactivities, and therapeutic potential of BCP and CA. Then, the relevance of mitochondrial dysfunction and NLRP3 inflammasome activation in microglia-mediated neuromodulation and neuroinflammation is briefly discussed. Finally, how BCP and CA, by modulating these mechanisms, may exert their neuroprotective and anti-inflammatory effects in chronic neurodegenerative disorders, such as AD, PD, and retinal degeneration, is proposed.

## 2. Oxidative Stress and Inflammation as Common Aspects in the Pathogenesis of Chronic Diseases: A General Overview

In response to injury/infection, acute inflammation supports the attentive orchestration of the innate and adaptive immune responses (IIR and AIR). However, the persistent and repetitive activation of the immune system can result in low grade inflammation. This chronic condition can disrupt multiple systems via systemic effects on the nervous system, and locally cause changes induced by cytokine expression throughout multiple bodily tissues.

The IRs are activated by the presence of stimulatory signals, including the pathogen-associated molecular patterns (PAMPs) and damage-associated molecular patterns (DAMPs), triggering monocyte/macrophage survival, and the consequent gene expression of pro-inflammatory factors and suppression of anti-inflammatory mediators [50]. Thus, Toll-like receptors (TLRs), DNA sensors such as cyclic GMP-AMP synthase (cGAS), and retinoic acid-inducible gene-I (RIG-I)-like receptors (RLRs), are identified as a significant class of recognition receptors (PRRs, pattern recognition receptors). Cytokine and chemokine networks also mediate the IIR and AIR, resulting in antagonistic, synergistic, and multiple effects [51].

Generally, activated macrophages, neutrophils, dendritic cells (DCs), and natural killer (NK) cells are actively engaged in the IIR [52,53]. Cytokines such as IL-1, IL-6, TNF-α, and interferon-alpha (IFN- α), as well as endogenous biological markers including cholesterol, insulin, and lipoproteins are critical in ensuring the effectiveness of IIR [15,54,55]. Unlike IIR, AIR generates immunological memory in which pathogens are “remembered” through memory B and T cells, thus by stopping an infection by the same pathogen before it can cause symptoms and preparing the body’s immune system for future challenges. Antigen presentation via an APC (antigen-presenting cell), such as a DCs or macrophage, stimulates T cells to become either “cytotoxic” CD8+ cells (T_C_), which directly lyse the antigen-bearing cells, or “helper” CD4+ cells (T_H_). In turn, T_H_1 effector cells have central roles in producing cytokines (IL-2, IFN-γ, and TNF-α), driving T_C_ activation, whereas T_H_2 cells are responsible for IL-4 and IL-5 production, which mediate the humoral immune response through the activation and differentiation of B cells [56]. Other T cells subsets, such as T regulators (T_REG_) and T_H_17, are important modulators of other immune responses. T_REG,_ by releasing IL-10 and transforming growth factor-beta (TGF-ß), down-regulates the excessive activation of T_H_1 and T_H_2, thus preventing allergies and autoimmune responses from taking place. T_H_17, for its part, produces IL-17, thus playing a central role in antimicrobial defences and the recruitment of neutrophils, particularly to mucous membranes.

Engagement of PRRs with their cognate ligands activates different intracellular signalling and nuclear transcription factors, including interferon-response factors (IRF), AP-1, and NF-κB. IRF3 and 7 promote the production of type I interferons (IFN-α and ß), leading to JAK-STAT pathway activation as well as the expression of ISGs (interferon-stimulated genes) [57]. NF-κB drives the expression of pro-inflammatory cytokines, such as TNF-α and IL-6, and the onset of inflammasome assembly [58,59]. The inflammasome is a multimeric platform of proteins involved in both initiation of inflammation and some types of cell death [60]. In this regard, the most extensively studied is the nucleotide-binding oligomerization domain-like receptor protein 3 (NLRP3) inflammasome, composed of NLRP3, a CARD-containing adaptor (ASC), and caspase-1 [61]. The NLRP3-activated caspase-1 cleaves pro-IL-1 and pro-IL-18 into their mature forms [59] and also interacts with propyroptotic factor gasdermin D (GSDMD) [62]. GSDMD facilitates IL-1β/IL-18 secretion and induces inflammation-associated cell death, also known as pyroptosis [63]. The IL-1β-mediated neutrophil recruitment to the inflammatory site aids in the elimination of viruses. However, the persistent/excessive inflammatory infiltration and pyroptosis-induced cell injury are responsible for tissue damage/immunopathology [64,65,66]. If this process is inefficient or prolonged, acute inflammation can progress to the chronic stage that is associated with many inflammatory diseases. Therefore, a direct association between the abnormal activation of NLRP3 and the pathogenesis of AD, obesity, multiple sclerosis, diabetic complication, inflammatory bowel diseases, and gout has been reported [67,68].

The generation of ROS and RNS has a central role in inflammatory disease progression. The ROS are produced by polymorphonuclear neutrophils (PMNs) and macrophages, which are directly involved in the host–defence response. The dramatic increase of ROS generation (also known as *oxidative burst*) leads to the alteration of membrane lipids, proteins, and nucleic acids, irreversible damage to the mitochondria and endoplasmic reticulum, as well as the promotion of endothelial dysfunction via oxidation of critical cellular signalling proteins, including tyrosine phosphatases. Protein oxidation causes the release of peroxiredoxin 2 (PRDX2) that acts as an inflammatory signal [69]. A recent study also demonstrated that ROS regulates the release of IL-1β by directly interfering with the NF-κB pathway [70]. In addition, the presence of elevated concentrations of superoxide anions (O2^−^) and nitric oxide (NO) in inflamed tissues facilitates the rapidly formation of RNS, especially peroxynitrite (ONOO^−^), which is the most cytotoxic metabolite [71]. Therefore, RNS-induced nitrosative stress adds to the mechanism of inflammation and tissue injury of ROS. Furthermore, chronic OS can cause the recruitment of immune cells and trigger the overexpression of NF-kB and AP-1, intensifying the inflammatory response and thus creating a chronic inflammatory condition [72,73,74]. In these conditions, stimulation of TGF-ß1, inducible nitric oxide synthase (iNOS), cyclooxygenase-2 (COX-2), and matrix metalloproteinases (MMPs) can drive inflammatory disease development [72,73,75,76,77,78]. Therefore, OS can trigger IRs, which in turn amplify OS [12,13].

## 3. β-Caryophyllene

### 3.1. Chemistry, Vegetable Sources, and Pharmacokinetics

BCP [C_15_H_24_] is a bicyclic sesquiterpene lactone compound chemically known as (trans-(1R,9S)-8-methylene-4,11,11-trimethylbicyclo (7.2.0) undecane. As reported in the Essential Oil Database [79,80], it is commonly found in the essential oils of many medicinal, ornamental, and edible plants (~2000) such as *Achyrocline satureioides* L., *Syzygium aromaticum* L., *Ocimum basilicum* L., *Cinnamomum species*, *Cannabis sativa* L., *Rosmarinus officinalis* L., *Teucrium* spp., and *Piper nigrum* L. (Figure 1). It has been evaluated safety by the Research Institute for Fragrance Materials (RIFM) and it has been approved by the Food and Drug Administration (FDA) and by the European Food Safety Authority (EFSA) as a flavouring, which can be used in food additives and cosmetic applications [81]. No toxic effects have been observed following oral administration of BCP in mice (2000 mg/kg) [82]. Moreover, literature data suggest its inclusion as an adjuvant into the dietary management of chronic lifestyle diseases [83]. BCP usually occurs in the form of trans-caryophyllene together with its minor isomers, including α-humulene, iso-caryophyllene, and its oxidative derivative BCP oxide (BCPo). In vivo studies have reported the bioavailability and pharmacokinetics of BCP after oral administration and inhalation [84,85,86]. In particular, the pharmacokinetic parameters of BCP and BCP/β-CD (β-cyclodextrin) inclusion complex were determined in rat plasma after oral administration at a dose of 50 mg/kg. Generally, the BCP/β-CD complex showed more favourable mean plasma concentration–time profiles as compared to free BCP, exhibiting earlier T_max_ (time to reach maximum plasma concentration, 2.80 ± 0.80 h), higher C_max_ (maximum plasma concentration, 0.56 ± 0.35 μg/mL), and better bioavailability [84]. In addition, the distribution of inhaled volatile BCP in murine organs and tissues, including the lung, brain, serum, heart, olfactory bulb, liver, epididymal fat, kidney, and brown adipose tissue has been demonstrated [86]. In this context, time-course analysis further revealed that BCP was detectable in liver, brain, and brown adipose tissue up to 24 h after exposure. BCP is a highly lipophilic compound (log p = 4.52), and this would explain its presence in high amounts in brown adipose tissue, as well as the possibility to cross blood–brain barrier (BBB), reaching the brain [87].

### 3.2. Therapeutic Potential and Polypharmacological Activities of β-Caryophyllene

Gertsch and colleagues [41], by using in vitro, in vivo, and in silico studies, first reported BCP as a full and selective agonist of CB_2_R (Ki = 155 ± 4 nM), a subtype of the G protein-coupled receptor (GPCR) of the endocannabinoid system (ECS). BCP intensively activates CB_2_R exclusively through hydrophobic interactions. In addition, it acts as a negative allosteric modulator (NAM) of CB_2_R binding with a pki of 2.37 µM [88]. This may provide pharmacological advantages with reference to improved target specificity and reduced/null off-target negative effects [89,90,91]. BCP does not produce psychomimetic effects due to the absence of binding affinity to the human CB_1_R, thereby opening the door to vast and new therapeutic applications [92]. In recent years, CB_2_R has been gaining more and more attention as a pharmaceutical target for many diseases. It is mainly expressed by immune cells, including B and T cells, T_C_, T_H_, macrophages, NK cells, neutrophils, basophils, eosinophils, mast cells, platelets, as well as microglia, dendritic cells, and astroglia [93], thus representing a specific target for immune regulation. In line with this, a strong immunomodulation exerted by CB_2_R activation [94,95], as well as BCP-dependent anti-inflammatory and antioxidant effects through activation of the cannabinoid receptor have been reported [42,96,97,98]. In particular, BCP exerts inhibitor effects on lipopolysaccharide (LPS)-induced ERK1/2 (extracellular signal-regulated kinases) and JNK1/2 (c-Jun N-terminal kinase) activation in macrophages [42]. BCP–CB_2_R interactions prevent LPS-induced oligodendrocyte toxicity and microglia imbalance [98,99], as well as C6 glioma cell excitotoxicity [100]. Moreover, a standardized non-psychotropic *C. sativa* extract containing BCP displays significant activity in reducing pro-inflammatory cytokine synthesis in activated BV-2 microglia cells through CB_2_R signalling [101]. In an experimental model of autoimmune encephalomyelitis, BCP was described as regulating local and systemic immunity [102]. In primary splenocytes, BCP exhibits immunomodulatory properties by inhibiting T_H_1/T_H_2 cytokines, including IL-2, IL-4, IL-5, IL-10, and IFN-γ [103]. The potential of BCP to improve antioxidant power has also been demonstrated in different organs, including brain, heart, intestine, liver, kidneys, stomach, pancreas, and blood [104]. Additionally, radical scavenging and ferric reducing antioxidant BCP properties, and the ability to inhibit lipid peroxidation and GSH depletion, have been reported [105,106].

Therefore, due to its pleiotropic properties (including anti-diabetic, antitumoral, and anti-allodynic activities), the CB_2_R-mediated therapeutic potential of BCP in the neuroprotection, cardioprotection, hepatoprotection, nephroprotection, and gastroprotection have been demonstrated in many studies [47,48,87,98,99,107,108,109,110,111,112,113,114,115,116,117,118,119,120,121,122,123,124].

An in silico study has also revealed the ability of BCP to target a syndrome coronavirus 2 (SARS-CoV-2) main protease (M^pro^; 3-chymotrypsin-like protease, 3CL^pro^) implicated in the translation of viral RNA into viral proteins [125]. In particular, BCP can interact with M^pro^ by establishing pie–alkyl interactions with PHE 294, showing good affinity (−7.2 kcal/mol) and drug-like properties. In line with this, Muthuramalingam and colleagues [126] have recently described BCP and BCPo as bioactive compounds capable of targeting COVID-19 immune genes implicated in different signalling pathways, extending the therapeutic potential of BCP to antiviral activity against COVID-19.

BCP–CB_2_R interactions trigger several intracellular signallings, including cAMP inhibition, intracellular calcium release, stimulation of SIRT-1/PGC-1α (sirtuin 1/peroxisome proliferator-activated receptor γ coactivator-1α), and AMPK (AMP-activated protein kinase) pathways, as well as mitogen-activated kinase (MAPK) activation [42,119,127,128,129]. Notably, MAPKs can modulate PPAR activity through direct phosphorylation [130]. PPARs (subtypes α, γ, and β/δ) are a subgroup of ligand-activated transcription factors, co-expressed in different types of organs and tissues, and playing a critical role in regulating energy production, lipid metabolism, inflammation, fibrosis, and immunity [131,132]. Activated PPARs can heterodimerize with the retinoid X receptor (RXR), thus altering the transcriptional activity of target genes. Generally, PPARs-mediated anti-inflammatory activity is based on the repression of NF-κB and Nrf2/CREB (nuclear factor erythroid 2 related factor 2/cAMP-responsive element-binding protein) signalling, and/or inhibition of macrophage activation [132]. This leads to a strong attenuation of pro-inflammatory cytokine and chemokine production, and the arrest of the transcriptional activity of other stress response elements, including COX2 and iNOS [131]. BCP-induced CB_2_R activation stimulates PPARγ activation via upregulation of PGC-1α, attenuating diet-induced metabolic and neurobehavioral disorders in rats [48]. Additionally, PPARγ is involved in BCP-mediated anti-arthritic and anti-inflammatory activities in mice [133]. In addition, the protective effects of BCP against LPS-triggered toxicity involve the CB_2_R-dependent activation of the Nrf2)/HO-1/antioxidant axis and PPAR-γ [98]. On the other hand, BCP can also bind and activate PPARα (equilibrium dissociation constants value, K_D_ = 1.93 µM) directly by interacting with the ligand-binding domain, thereby modulating cellular intracellular triglyceride as well as fatty acid uptake and oxidation [134]. In addition to the activation of CB_2_R and PPAR, BCP also interacts/binds to other receptors, such as TLR [96,135,136], TRPV [137], µ-opioid [138], and histaminergic [139]. Moreover, BCP inhibits NLRP3 inflammasomes [136] and various enzymatic activities, including lipase, amylase, reductase, α-glucosidase, acetylcholinesterase, and secretase. Of note, in rat and human liver in vitro, the inhibitory effects of BCP, α-humulene, and BCPo toward enzymes involved in xenobiotic detoxification and metabolism (cytochromes P4503A, CYP3A) have been reported [140].

## 4. Carnosic Acid

### 4.1. Chemistry, Vegetable Sources, and Pharmacokinetics

CA (C_20_H_28_O_4_) is a polyphenolic compound that belongs to the large family of terpenoids (a natural benzenediol abietane diterpene) [141]. It is mainly found in the edible plants salvia (*Salvia officinalis* L.) and rosemary (*Rosmarinus officinalis* L.) (Figure 1) [142], particularly in the leaves, sepals, and petals [49,143,144]. It can penetrate the BBB and pile up in the mammalian brain tissue [145,146], reaching levels of 4.8 μmol/kg tissue after 1 h from oral administration of CA (3 mg), and 2.4 μmol/kg tissue after 3 h in mice. In vivo experiments have estimated the CA bioavailability (40.1% after 360 min) after oral (CA at 64.3 ± 5.8 mg/kg) and intravenous (20.5 ± 4.2 mg/kg) administration in rats [147]. In this context, CA was distributed in plasma, liver, intestine, and muscle. In line with this, Romo Vaquero and colleagues [146] reported the distribution of CA and its derivatives in different organs, including the small intestine, colon, caecum, liver, and brain after oral administration with CA-enriched rosemary extract (40%, *w*/*w*) in female Zucker rats. In this regard, plasmatic CA levels reached 26.6 μM at 0.4 h, and in brain it was found at 1.5–1.9 μg/g after a subchronic (64 days) administration.

### 4.2. Therapeutic Potential, Biological Activities, and Pharmacological Effects of Carnosic Acid

CA is a safe pro-electrophilic drug (PED) that in response to oxidation is converted to its electrophilic form. Therefore, given the nature of its chemical structure (characterized by the presence of the so-called catechol moiety), CA can act as a ROS scavenger (direct effect), protecting biomembranes from oxidation [148,149,150]. CA may be thought of as a pathologically activated therapeutic (PAT) drug [151,152]. It can exert tissue protection through its active electrophilic state, triggered only by ROS in a given oxidised/inflamed tissue (as occurs in neurodegenerative diseases). As an electrophilic agent, CA can interact and activate the Keap1/Nrf2/ARE transcription pathway (indirect effect), triggering the synthesis of endogenous antioxidant “phase 2” enzymes, including HO-1, glutathione S-transferase (GST), γ-glutamyl cysteine synthetase (γ-GCS), thioredoxin reductase (TrxR), and NQO1 as well as GSH and anti-inflammatory enzymes [145,151,153]. Therefore, by activating the Keap1/Nrf2/ARE transcription pathway, CA can exert neuroprotection in different model systems [145,154,155,156,157,158]. Through the same molecular mechanism, CA protects from LPS-induced liver damage [159,160] and ROS/RNS generation in RAW 264.7 macrophages [161,162]. In addition, CA-induced antioxidant effects and up-regulation of mitochondrial GSH in SH-SY5Y cells are associated with the activation of the PI3K/Akt/Nrf2 axis [153,163,164,165], confirming a role for CA as a neuroprotective agent. CA protects rat cerebellar granule neurons from the in vitro modelling of nitrosative stress and caspase-dependent apoptosis by activating a PI3K pro-survival pathway [166]. The effectiveness of CA as an anti-inflammatory compound is also due to its ability to inactivate NF-kB, ERK/JNK/p38 MAPKs, and fork-head box protein O3a (FoxO3a) signalling pathways [167,168,169,170,171,172]. In particular, CA has the ability to reduce myeloperoxidase and iNOS levels along with NLRP3 and CASP1 expression in male BALB/c mice [171]. Concurrently, it increases levels of Nrf2, GSH, and SOD. In addition, CA has been described to suppress NLRP3/NF-kB, PI3K/AKT, and sterol regulatory element binding protein 1 (SREBP-1c) pathways [173].

Therefore, to date, several studies have shown the antioxidant properties [49] and anti-inflammatory role of CA in distinct in vitro and in vivo models, including mouse hyperalgesia, carrageenan-induced collagen-induced arthritis, dextran sulfate sodium-induced acute colitis, and Receptor Activator of Nuclear Factor-κB ligand (RANKL)-induced osteoclastogenesis [169,170,171,174]. In addition, therapeutic effects have been declared in preclinical in vivo models of inflammation, such as acute liver and lung injury, non-alcoholic fatty liver disease injury, cirrhosis, cardiotoxicity, diabetes and hepatic fat accumulation, brain injury, and hepatocarcinoma [170,173,175,176,177,178,179,180,181,182,183,184].

## 5. Neuroprotective Potential of β-Caryophyllene and Carnosic Acid in CNS and Visual System

### 5.1. Mitochondria as a “Trait-D’Union” in the Complex Interplay between Neuroinflammation and Neurodegeneration

The central nervous system (CNS) can be affected by a wide range of neurological disorders (NDD), such as AD and PD, which are common causes of morbidity and mortality worldwide [185]. AD causes brain cognitive impairment and dementia, being characterized by OS, defective cholinergic neurotransmission, extracellular β-amyloid (Aβ) plaque, and intracellular neurofibrillary tangles (NFTs), forming hyperphosphorylated tau-protein (pTau) deposited in human brain tissues [186,187]. In this regard, the increased levels of methylglyoxal (MG)-derived advanced glycation end products (AGEs) may also be responsible for cell damage and inflammation [188,189].

For its part, the pathology of PD typically exhibits three cardinal features, such as α-synuclein (α-Syn) aggregation (named Lewy bodies), neuroinflammation, and dopaminergic neurodegeneration in the *substantia nigra pars compacta* (SNpc), resulting in resting tremors, along with bradykinesia and muscle rigidity. A growing body of evidence indicates that impairment of the ubiquitin–proteasome system (UPS) is associated with the aetiology of PD. Interestingly, beyond its function in the UPS, the parkin-mediated modulation of the TGF-β signalling pathway (ARTS), a mitochondrial pro-apoptotic protein, and X-liked inhibitor of apoptosis protein (XIAP), a key member of the newly discovered family of intrinsic inhibitors of apoptosis (IAP) proteins, plays a crucial role in preventing PA. Parkin acts as an E3-ubiquitin ligase to reduce the ARTS levels through UPS-mediated degradation [190]. Over-expression of parkin protects cells from apoptosis induced by ARTS. Therefore, mutations of parkin results in impairment in the UPS machinery, leading to accumulation of misfolded proteins, increased levels of ARTS, and progressive neuronal degeneration and cell death. Notably, the marked presence of misfolded proteins can also generate ROS overproduction, which can lead to NLRP3 inflammasome activation, and the subsequent release of pro-inflammatory cytokines from microglia, contributing to neuronal death [191,192,193,194].

In accordance with their multifaceted roles in cellular metabolism and signalling and cell death, mitochondria are essential for maintaining neuronal integrity and coherence [195]. Consistent with their high energy requirements, neurons are especially vulnerable to mitochondrial dysfunctions [196,197]. Loss of mitochondrial homeostasis and function, or impairment of mitochondrial quality control (e.g., mitophagy), is a cause or consequence of neurological diseases [198]. On the other hand, parkin and PINK1 are PD-associated genes and related to mitophagy [199]. In addition to loss of mitochondrial homeostasis, others pathogenetic aspects, such as neuroinflammation and activation of microglia, excitotoxicity, and OS, are frequently described in the progression of neurodegenerative diseases [200,201,202]. In particular, neuroinflammation and mitochondrial dysfunction never work alone in NDD, sharing multiple and intricate molecular links responsible for neuronal death and the aggravation of the disease.

Glial cells, including microglia (the brain-resident phagocytic cells), oligodendrocytes, and astrocytes, are central to neuronal health and functionality. Astrocytes regulate the architecture and activity of neuronal circuits, showing remarkable contributions to synapse formation, maintenance, and pruning. Microglia play a crucial role in the neurobiological and neurophysiological homeostasis of the CNS (e.g., elimination of dead neurons and cell debris, and remodelling synapses). In response to pathological conditions, the microglia phenotype changes from *surveillant* to pro- (M1 cells) or anti-inflammatory (M2 cells: M2a, implicated in repair and regeneration; M2b, involved in immune-regulatory activity; M2c, an acquired-deactivating phenotypes) [203,204]. However, depending on the surrounding milieu, microglia can also acquire multiple states/phenotypes, revealing a much higher complexity [205,206]. By releasing different molecules, microglia and astrocytes establish bi-directional communication, functioning as a single coordinated unit. Therefore, their mutual influence also contributes to increase the level of complexity of the microglia phenotype spectrum, especially in the context of maintaining neuroinflammatory homeostasis. Microglia are clearly engaged in neuroinflammation by cooperating with other immune cells and by producing a plethora of cytokines and endogenous lipids, including pro-resolving mediators and eCBs.

Being the CNS resident immune cells, they are equipped to detect PAMPs, DAMPs, and resolution-associated molecular patterns (RAMPs) through their vast array of PRRs. PRRs, such as CR3 and TLR-4, can detect “danger” signals in the surrounding milieu; phlogistic soluble factors, cellular debris and bacterial components (e.g., LPS), IL-1, IL-8, IFN and TNF, nucleic acids, ATP, and misfolded proteins can alter CNS homeostasis. Therefore, the specific recognition and internalization of these ligands leads to the microglia activation toward a pro-inflammatory-M1-phenotype characterized by the activation of NLRP1/3 inflammasomes and, consequently, the release of pro-inflammatory and immunomodulatory mediators (e.g., IL-1β, IL-6, IL-23, and TNF), chemokines, and the induction of NADPH oxidase- and iNOS-triggered OS, which may results in excito- and neuro-toxicity, damage to the myelin layer, and, finally, dysfunction and death of neurons [25,26,27] (Figure 2).

Interestingly, similar to macrophages, dendritic cells, and cancer cells, M1/M2 transition requires significant changes in metabolic pathways [207,208,209]. In this regard, mitochondrial metabolism governs and regulates microglial responses through a phenomenon that has been termed immunometabolism [210,211]. If on the one hand the engagement of increased glycolysis and pentose phosphate pathway activation is required for microglia pro-inflammatory phenotype, on the other hand, the surveillance (homeostatic) and anti-inflammatory states are sustained by efficient mitochondrial function and oxidative phosphorylation [212] (Figure 2). In supporting these metabolic shifts, microglia rearrange their redox system. Therefore, mitochondrial homeostasis and metabolic reprogramming are fundamental and integral aspects in dictating phenotypic responses for changing the course of neuropathophysiology. In this sense, a key role has been shown for ROS, antioxidant pathways, and the transcriptional regulator Nrf2 [213]. The fire of neuroinflammation is also fuelled by mitochondrial fragments that drive microglia activation and propagate inflammatory neurodegeneration from microglia to astrocytes (A1 polarization, pro-inflammatory/neurotoxic phenotype) [214,215].

In this regard, mitochondria have emerged as important hubs for signalling pathways implicated in innate and adaptive immune responses (the so-called mito-inflammation), representing a direct connection between neuroinflammation and neurodegeneration [216,217]. On the one hand, they can act as intracellular signalling pathways downstream of exogenous PAMPs. On the other, they regulate the inflammatory polarization of immune cells acting as a source of mitochondrial-derived DAMPs (mtDAMPs) after injury and increased membrane permeability. Therefore, misfolded Aβ, α-Syn, and pTau, which predominate in AD and PD, can induce mitochondrial dysfunction in neurons [218,219,220,221,222]. In turn, their excessive release into the extracellular space causes a ripple effect, promoting further mitochondrial damage in surrounding neurons and glial cells (Figure 3 and Figure 4). In particular, a cell-to-cell transmission proposed mechanism of α-Syn states that it can self-propagate among cells in a prion-like fashion, contributing to widespread neurodegeneration [223].

Therefore, mtDAMPS are recognized by immune receptors of microglia and exacerbate neuroinflammation. Indeed, due to the endosymbiotic origin of mitochondria, mtDAMPs have high immunogenic potential and are improperly recognized by the immune system as markers of infections [224]. Under stress conditions, the release of mtDAMPs, such as mtROS, mtDNA (particularly oxidized mtDNA), cardiolipin, ATP, transcription factor A mitochondria (TFAM), and mtCa^2+^, mediates the expression and release of numerous pro-inflammatory mediators in a cell-intrinsic or cell-extrinsic manner (Figure 3 and Figure 4). In the cytoplasm, mtDNA can activate the NLRP3 inflammasome, inducing the caspase-1-dependent expression of IL-1β and IL-18 [225]. Additionally, mtDNA is sensed by the cGAS-cGAMP-STING (cyclic GMP-AMP-Stimulator of Interferon Genes) pathway, which activates TBK1-IRF3-IFN (tank-binding kinase 1- interferon regulatory factor 3-interferon) signalling, thus triggering the expression of inflammatory mediators [226]. Otherwise, mtDNA in the extracellular milieu is recognized by TLR9 (via unmethylated CpG-rich DNA) located on the surface of the endosome of other glia, leading to the activation of the NF-κB signalling pathway. Similarly, mtROS and cardiolipin released in the cytoplasm act upstream of the NLRP3 inflammasome activation. In this regard, one proposed mechanism for ROS-mediated NLRP3 activation would involve the diffusion of mitochondrial antiviral signalling protein (MAV) from mitochondria to the cytoplasm [227,228].

Moreover, the ATP and TFAM released from neuron/glia extracellularly promote proinflammatory microglial activation by interacting, respectively, with the purinergic P2X7 receptor (inducing the activation of NLRP3 inflammasome) and advanced glycation end-products (RAGE) or TLR4 (upregulating the levels of NF-κB) on the membrane surface of glia [226].

Therefore, depending on cell type-specific differences, mitochondrial damage can contribute to the innate immune responses, essentially relying on two major programs, the cGAS/STING pathway and NLRP3 inflammasomes [229,230,231,232,233]. In turn, activated glial cells release inflammatory factors, triggering cellular cascades, which can exacerbate mitochondrial damage, triggering and intensifying a vicious circle between neuroinflammation and mitochondrial dysfunction, which finally results in NDD. Notably, mitochondrial quality control by mitophagy is a crucial intrinsic response to prevent neuroinflammation, by limiting mtDNA and mtROS release from damaged mitochondria, and NLRP3 inflammasome activation [234,235,236,237].

### 5.2. CA and BCP as a Promising Immunomodulatory Strategy to Maintain Glial and Mitochondrial Homeostasis

CA and BCP can readily cross the BBB and accumulate in brain regions. In this regard, CA can also give rise to an active electrophilic form at sites of OS. Therefore, growing evidence suggests that both compounds may exert neuroprotective effects by preventing mitochondrial dysfunction and inhibiting activated microglia, substantially through the activation of the CB_2_R/PPARγ axis and Nrf2/HO-1/NQO1 signalling and, at least in part, via inhibition of NLRP3 inflammasome activity (Figure 3 and Figure 4). Mitochondrial quality control has a unique role in maintaining a healthy mitochondrial population in neurons. In this regard, CA-mediated neuroprotection through the activation of pro-survival signalling pathways, including regulation of apoptosis and autophagy, and molecular mechanisms related to mitophagy, mitochondrial dynamics, and mitochondrial biogenesis has also been shown [168,238,239,240,241] (Figure 3). Therefore, results from different in vitro/in vivo experimental models of PA and AD reported the beneficial effects of BCP and CA, suggesting that their treatment may be a promising strategy in the management of neurodegenerative diseases.

### 5.3. CA Drives the Shift of Microglia Polarization in Neuroinflammation and Ameliorates Neurodegeneration in Models of AD and PD

In LPS-activated mouse microglia (MG6 cells), CA can inhibit the release of NO, IL-1β, and IL-6, and reduce iNOS expression levels [159]. CA have also the ability to activate the Nrf2–HO-1 antioxidant axis, suppressing the TNF-α signalling pathway and modulating the inflammation in BV2 microglia cells challenged with LPS or INF-γ [242]. In line with this, the combination of CA with different omega-3 free fatty acid preparations, synergistically regulates the inflammatory response of BV-2 microglia cells, attenuating the LPS/IFNc-induced pro-inflammatory phenotype and enhancing anti-inflammatory responses [160]. In this regard, the presence of the combination reduces the levels of inflammatory mediators, including PGE2, IL-6, superoxide, and NO; prevents iNOS activity and COX-2 upregulation; and increases secretion of the anti-inflammatory cytokine IL-10, promoting a shift toward the anti-inflammatory M2 phenotype. CA may exert antioxidant and anti-inflammatory responses as well as neuroprotective effects via the inhibition of NLRP3 inflammasome activity related to the upregulation of the Keap1/Nrf2/ARE transcription pathway, and the subsequent induction of “phase 2” enzymes [243]. In particular, α-Syn- and/or Aβ-induced IL-1β release in human-induced pluripotent stem cell (hiPSC)-derived microglia (hiMG) is inhibited by CA, suggesting a significant reduction of the NLRP3 inflammasome activation. In human, αSyn–antibody complexes are neuroinflammatory given their ability to increase inflammasome-mediated IL-1β secretion [192]. Notably, in the presence of αSyn–antibody complexes, CA suppresses the release of IL-1β from hiMG, indirectly indicating NLRP3 inflammasome inhibition [243]. The upregulation of CEBPβ expression, a member of the CCAAT-enhancer-binding protein (CEBP) family, is associated with the increased expression of NFkB target genes, resulting in glia-mediated neuronal damage [244,245,246].

Oral administration of nanocarrier-packaged CA inhibited CEBPβ transcriptional activity and reduced its interaction with NFkB in an APP/PS1 double transgenic AD mouse model, exhibiting amyloidosis and neuroinflammation. This resulted in attenuation of microglial and astrocyte activation, mitigation of neuroinflammatory response, and inhibition of Aβ secretion [247]. These findings are in line with other studies, demonstrating the potential for promoting the beneficial effects of AC on neurodegeneration in various in vivo and in vitro models of AD and PD. In an experimental rat model of AD, Azad and colleagues [248] have shown that the administration of CA (at a dose of 3 mg/kg) reduces hippocampal neuronal cell death, thus preventing Aβ-induced neurodegeneration. In the same experimental model, the ability of CA to ameliorate the progressive cognitive decline induced by Aβ, in terms of spatial and learning memory, was not demonstrated [249]. Later, Lipton and colleagues [250] have reported the CA-mediated neuroprotection in in vitro (in rat primary cortical neurons) and in vivo (in two transgenic mouse models, hAPP-J20 and 3xTg AD) models of AD, showing its ability to rescue different AD-related phenotypes. In particular, CA treatment for 3 months (trans-nasal administration) improved impairment in learning and ameliorated hippocampal damage in mice. Histological analysis also revealed that CA decreases synaptic loss, astrogliosis, deposition of Aβ peptide, and accumulation of pTau in the hippocampus. Moreover, in neurons exposed to oligomeric Aβ, CA reduces dendritic spine loss.

In SH-SY5Y human neuroblastoma and U373MG astrocytoma cells, CA treatment lowers the production of Aβ40/42 and Aβ40/42/43, respectively, by increasing the expression of α-secretase TACE (tumour necrosis factor-α-converting enzyme) [251,252]. Moreover, in the same experimental model, CA pre-treatment (10 µM for 1 h) partially reduced the Aβ42/43-induced apoptosis by suppressing the activation of caspase activation pathways and reducing the presence of cleaved PARP [253]. As described above, MG is a potent precursor of AGEs and is closely associated with an increase of OS and inflammatory status in AD [254,255]. Interestingly, in SH-SY5Y cells, CA exhibits neuroprotection against MG-mediated neurotoxicity, directly activating the PI3K/Akt/Nrf2 signalling pathway [163]. *Caenorhabditis elegans* has emerged as an important in vivo model in the context of AD research due to its ease of obtaining stable transgenic models that recapitulate many key aspects of protein aggregation in humans [256,257,258]. CA promotes the health span of *C. elegans* by delaying Aβ-induced paralysis and increasing the expression of SOD-3 [259]. Later, the same research team demonstrated that CA attenuates Aβ42 oligomerization and deposition in AD transgenic *C. elegans* and markedly protects neurons from Aβ-induced toxicity and cholinergic dysfunction [260].

In the last few years, the elaboration of in vitro/in vivo models of PD has made use of various neurotoxic agents. Therefore, the dose-dependent protective effects of CA against dieldrin-induced neurotoxicity in cultured dopaminergic cells (SN4741) have been demonstrated [261]. Similarly, CA treatment attenuates SH-SY5Y cell death induced by exposure to 6-hydroxydopamine (6-OHDA, a selective catecholaminergic neurotoxin). In this regard, CA-mediated neuroprotection is related to Nrf2-driven expression of GSH, resulting in the down-regulation of JNK and p38 signalling pathways [262]. In the same cell type, CA affords neuroprotection against 6-OHDA-mediated neurotoxicity by activating the PI3K/Akt/NF-κB pathway, resulting in induction of GSTP (glutathione S-transferase-P) protein expression [263] and by upregulating parkin protein expression and the activity of the ubiquitin–proteasome system (UPS) [238]. In this regard, it has been further shown that the CA-induced expression of parkin results in UPS-mediated reduction of the ARTS levels and subsequent XIAP-dependent decrease of 6-OHDA-induced apoptosis in SH-SY5Y cells [264]. CA also exerted neuroprotective effects against cyanide-associated brain injury in a non-Swiss albino (NSA) mouse model of cyanide poisoning, and in vitro in hiPSC-derived neurons through activation of the Nrf2/ARE transcriptional pathway [265]. Consistent with these findings, in a 6-OHDA rat model of PD, CA ameliorated the motor performance, inhibited OS, and reduced apoptosis [266].

Interestingly, activation of the CA-mediated neuroprotective mechanism on levodopa (L-dopa)-induced dyskinesia (LID) (commonly seen in PD patients treated with L-dopa) by in vivo and in vitro studies has recently been reported [267]. In an in vivo study, CA mitigates the LID-induced behavioural changes in 6-OHDA-treated rats by reversing the dopamine (D1-receptor) D1R-mediated activation of DARPP-32 and ΔFosB (two factors responsible for causing LID) and reducing L-dopa-dependent phosphorylation of ERK1/2 and c-Jun Ser63. Furthermore, in SH-SY5Y cells, CA prevents L-dopa-induced apoptosis by increasing c-Jun ser73 activation and decreasing the ERK1/2-c-Jun ser63 pathway. These effects were also associated with the induction of parkin expression, thus preventing the accumulation of ubiquitinated protein and D1R abnormal trafficking [267].

### 5.4. Mechanisms of CA-Activated Neuroprotection by Modulating Mitochondrial Homeostasis

CA has recently been described as a significant mitochondrial protective agent. According to its cytoprotective functions, Nrf2 have crucial roles in mitochondrial function and integrity [268], supporting the importance of the CA-induced Nrf2/HO-1/NQO1 signalling pathway in protecting mitochondria. On the other hand, PINK1/parkin-mediated mitophagy and PGC-1α-regulated mitochondrial biogenesis, as CA-related mechanisms aimed at preserving or renewing mitochondrial homeostasis, have also been reported (Figure 3). Generally, the importance of CA in the context of mitochondrial protection has been observed in in vivo and ex vivo experimental models. Moreover, this ability has also been studied using human SH-SY5Y neuroblastomas as an in vitro model of dopaminergic neurons.

### 5.5. In Vitro Experimental Models

In SH-SY5Y cells exposed to different chemical stressors, including 6-OHDA [262], MG [163,269], paraquat [153], H_2_O_2_ [165], and chlorpyrifos [164], CA improves mitochondrial functions, especially the activities of tricarboxylic acid (TCA) cycle enzymes and the electron transport chain (ETC), modulates the redox state, and inhibits the apoptosis system. In this regard, CA exhibits anti-apoptotic effects by acting via the mitochondria-dependent pathway involving antioxidant and bioenergetic effects that are associated, at least in part, with the activation of the PI3K/Akt/Nrf2 signalling pathway. In particular, pre-treatment with CA at 1 μM for 12 h prevents toxicant-induced increased levels of markers of mitochondria-related apoptotic cell death (BAX content, cytochrome c release, caspases-9 and -3 activity, cleavage of PARP, and fragmentation of DNA), and lipid peroxidation, protein carbonylation, and protein nitration in mitochondrial membranes [153,163,164,165,269]. Additionally, CA restores the activity of complexes I and V, as well as the levels of adenosine triphosphate (ATP), and prevents loss of mitochondrial membrane potential (ΔΨ_m_) [164,269], and the inhibition of enzymes involved in the maintenance of the bioenergetics state, including aconitase, α-ketoglutarate dehydrogenase, and succinate dehydrogenase [165]. Moreover, CA alleviates ROS and RNS, upregulates the contents of GR, GPx, glutathione-S-transferase, and Mn-SOD, and increases GSH synthesis by enhancing the γ-GCL activity [163,269]. The CA-induced mitochondria-related anti-apoptotic and antioxidant effects may involve the PI3K/Akt/Nrf2/HO-1 signalling pathway, since inhibition of the PI3K/Akt axis, Nrf2 silencing with small interfering RNA (siRNA), or inhibition of HO-1 by ZnPP IX abrogates the CA-mediated beneficial effects. Finally, in MG-challenged cells, CA may promote mitochondrial protection by a PI3K/Akt/Nrf2/γ-GCL/GSH axis, since inhibition of the PI3K/Akt axis and Nrf2 silencing suppresses the CA-promoted cytoprotection and the synthesis of GSH [269].

CA is also effective in promoting mitochondrial protection by upregulating the expression of NQO1 by the Nrf2–ARE antioxidant response pathway [270]. NQO1 is the main intracellular enzyme catalysing idebenone reduction, a synthetic quinone that, once reduced to idebenol, can bypass mitochondrial Complex I defects by donating electrons to Complex III. Therefore, it is used clinically to treat the Complex I disease Leber’s hereditary optic neuropathy (LHON). However, the clinical efficacy of idebenone depends on the intracellular presence of NQO1; otherwise, it can impair neuronal mitochondrial respiratory capacity by inhibiting Complex I-dependent respiration. CA exhibits mitoprotective potential in synergism with idebenone by stimulating NQO1-dependent mitochondrial bioenergetic rescue in COS-7 cells that express low endogenous levels of the enzyme but are competent to activate the Nrf2–ARE pathway [270].

In SH-SY5Y cells, CA alleviates 6-OHDA-induced neurotoxicity via activation of the UPS and the autophagy pathway, restoring the impairment of parkin protein, and increasing the parkin–beclin-1 interaction, respectively [238,239]. The central role of PINK1/parkin-mediated mitophagy in preserving neuronal health during the exposure to different challenges has been described [199,271]. In this regard, the protective role of CA in preventing 6-OHDA-elicited mitochondrial impairment in SH-SY5Y cells is related to its ability to activate the PINK1/parkin axis to ubiquitinate the voltage-dependent anion channel 1 (VDAC1) and enhance mitophagy [240]. The neuroprotection of CA in counteracting the toxicity of 6-OHDA also occurs through the modulation of mitochondrial dynamics and upregulation of the fusion protein optic atrophy 1 (OPA1), mediated by parkin/IKKγ/p65 pathway activation, to prevent released cytochrome c and apoptotic-related protein activation [272].

The parkin-interacting substrate (PARIS) plays an important role in the reduction of mitochondrial mass and activity by repressing the transcriptional expression of proliferator-activated receptor gamma coactivator-1-alpha (PGC-1α) [273,274]. It also contributes to the pathologic α-synuclein-induced degeneration, thus accelerating the loss of dopamine neurons. In this regard, parkin-mediated ubiquitination of PARIS and subsequent degradation through the UPS prevents the repression of PGC-1α, allowing mitochondrial biogenesis to take place [273]. Interestingly, the exposure of SH-SY5Y cells to 6-OHDA upregulates PARIS content, decreases PGC-1α levels, and reduces the mitochondrial biogenesis proteins NRF1 (nuclear respiratory factor 1) and TFAM (mitochondrial transcription factor A) [241]. In this context, CA pre-treatment reverses the 6-OHDA-induced effects and prevents the degeneration of dopaminergic neurons. Therefore, CA-induced parkin activation rescues the impaired mitochondrial biogenesis and promotes neuronal health by modulating the PARIS/PGC-1α axis [241].

### 5.6. In Vivo and Ex Vivo Experimental Models

In young adult CF-1 mice, the administration of CA at 1 mg/kg (48 h before isolation of mitochondria from the cerebral cortex) induced mitochondrial protection from subsequent 4-hyrdoxynonenal (4-HNE) exposure ex vivo [275]. In particular, in cortical tissue, CA attenuated the inhibition of mitochondrial complex I and II induced by 4-HNE and decreases the levels of 4-HNE-bound protein. Moreover, CA significantly enhanced the mRNA levels of the Nrf2–ARE-mediated gene HO-1. CA-mediated mitochondrial protection was also described in an experimental model of TBI (traumatic brain injury) in mice [276]. In particular, CA administration (1–3 mg/kg) at 15 min post-TBI decreased cortical lipid peroxidation, cytoskeletal breakdown markers, and protein nitration in samples obtained from the mice cerebral cortex at 48 h post-injury. Furthermore, CA rescued mitochondrial function and reduced oxidative damage to mitochondrial proteins. In mice exposed to cyanide, an inhibitor of mitochondrial electron transport, the administration of CA at 25 mg/kg.day^−1^ mitigated neuronal loss in different brain regions, including the neocortex, the hippocampus, as well as the striatum, and it restored synaptic density [265]. Due to the central role played by mitochondria in the neuronal structure and function, there may be a direct link between CA-mediated maintenance of the synaptic density and mitochondrial protection [277,278].

### 5.7. BCP and Its Neuroprotective Functions: CB_2_R-Mediated Regulation of the Glia-Mediated Neuroinflammation and Improvement of Neurodegeneration in Model of AD and PD

In a murine model of PD, pre-treatment with BCP prevented (1-methyl-4-phenyl-1,2,3,6-tetrahydropyridine) MPTP-induced neurotoxic injury of dopaminergic neurons, resulting in a significant inhibition of SNpc neuron loss and suppression of the autochthonal microglia and astrocytes, thereby reducing the release of pro-inflammatory cytokines within the surrounding tissues [279]. In a transgenic APP/PS1 mouse model of AD, the PPARγ-mediated anti-inflammatory effect of BCP was ascertained; in particular, BCP administration reduced the β-amyloid burden in the cerebral cortex and hippocampus underlying the cognitive impairment, thereby ameliorating the AD functional deficit [122]. In this context, BCP was also demonstrated to be effective in reducing microglial activation and astrogliosis, along with lower mRNA and protein levels of COX2 and some proinflammatory cytokines expression, such as TNF-α and IL-1β, in the cerebral cortex tissue. The administration of AM630 (CB_2_R antagonist) or GW9662 (PPARγ antagonist) compounds confirmed the PPARγ-mediated CBR_2_-dependent neuroprotective effect of BCP. Recently, an interesting study on primary microglial cells demonstrated that BCP treatment can promote a shift in microglial phenotypes, tilting the ratio of LPS-induced M1/M2 imbalance towards the M2 anti-inflammatory phenotype, also reducing the M1 counterpart [99]. In this regard, the reduction of proinflammatory mediators, such as prostaglandins and cytokines, along with a stimulation of anti-inflammatory molecules (e.g., IL-10, urea, and arginase-1) and antioxidant mediators (e.g., GSH), are implicated in the M1/M2 switch. Interestingly, low BCP doses promote anti-inflammatory responses through CB_2_R activation, while higher concentrations of the compound also result in PPARγ activation. Similarly, in an in vitro model of oligodendrocytes, BCP protected against LPS-induced neurotoxicity, lowering the global inflammatory and oxidative stress-related status via activation of the CB_2_R/PPARγ axis and Nrf2/HO-1 signalling [98]. It also ameliorated Aβ1–42-induced neuro-inflammation in BV-2 microglial cells through the suppression of TLR4, NO, PGE2, iNOS, and COX-2 expression, and the secretion of pro-inflammatory cytokines [96]. Recently, Zhang and colleagues [280] have shown that BCP may protect SH-SY5Y cells against Aβ neurotoxicity via the inhibition of JAK2 expression. BACE1 protein is crucial for the synthesis of amyloid precursor protein (APP). In this regard, activation of the “JAK2-STAT3-NF-kB-BACE1” pathway leads to Aβ oligomerization and deposition, as well as neurotoxicity [281]. Therefore, in PC-12 cells overexpressing the APP, BCP exerts neuroprotective effects through the inhibition of the “JAK2-STAT3-BACE1” signalling pathway [282]. In the MPTP-induced mouse model of PD, pre-treatment with BCP protected against neurotoxic damage of dopaminergic neurons by increasing the expression and activity of NQO1 [283]. The action of NQO1 prevents the toxic accumulation of dopamine quinones and therefore the inhibition of mitochondrial complex I and ROS overgeneration. In a rat model of acute amyloid toxicity, the *Pinus halepensis* essential oil (PNO, containing BCP) attenuated Aβ1-42-induced OS and memory impairment in rat hippocampus [284]. In particular, PNO administration for 21 days resulted in the suppression of acetylcholinesterase (AChE) activity, enhancement of antioxidant markers (e.g., SOD, CAT, GP_x_, and GSH), and reduction of Aβ-induced increase of malondialdehyde (MDA) levels. The potential neuroprotective properties of essential oils derived from leaves of *Aloysia citrodora* Palau (the major constituent is BCP) against OS and Aβ-induced neurotoxicity was also investigated [285]. Moreover, in a mouse model of PD, the essential oil of *Eplingiella fruticose* (containing BCP) complexed with β-cyclodextrin exhibited neuroprotective effects when administered at a dose of 5 mg/kg, p.o., for 40 days [286]. In this sense, it decreased membrane lipid peroxide levels and preserved dopaminergic depletion in the striatum and SNpc. The BCP neuroprotective mechanisms are summarised in Figure 4.

### 5.8. Neuroprotective Role of BCP Via Mitochondrial Homeostasis Management

BCP-mediated neuroprotective activity may be partly exerted through mitochondrial homeostasis and pathway modulation (Figure 4). Assis and colleagues [100] showed the ability of BCP to ameliorate glutamate-induced cytotoxicity in C6 glioma cells. In this regard, BCP reduces ROS and NO levels, and improves the cellular GSH antioxidant system via Nrf2 activation, partially dependent on CB_2_R activation. Most important, BCP re-establishes ΔΨ_m_. The possible involvement of CB_2_R in the antioxidant effects of BCP against (1-methyl-4-phenylpyridinium) MPP+-induced neurotoxicity in SH-SY5Y cells has been described [97]. In particular, BCP suppresses apoptosis and ROS generation, increases GSH and GP_x_ activity, and improves mitochondrial bioenergetic activity, restoring ΔΨ_m_. Exposure of rats to the complex I inhibitor rotenone replicated features of PD, including selective nigrostriatal dopaminergic degeneration, the activation of astrocytes and microglia, generation of OS, and the upregulation of proinflammatory cytokines and inflammatory mediators. In a rat model of PD (rotenone-induced PD), BCP administration (once daily for 4 weeks at a dose of 50 mg/kg body weight) improved the antioxidant system, restoring antioxidant enzyme activity (e.g., SOD and CAT), and inhibiting GSH depletion and lipid peroxidation. This resulted in attenuation of microglial and astrocyte activation, the rescuing of dopaminergic neurons, and mitigation of the neuroinflammatory response [287]. In the same experimental model, in a similar way, Javed and colleagues [112] demonstrated the CB_2_R-mediated neuroprotective activity of BCP. The clove oil obtained from *Syzygium aromaticum* (L.) Merr. and L.M. Perry is well known to contain BCP. In rats, the treatment with clove oil (0.05 mL/kg and 0.1 mL/kg) afforded neuroprotection against intra-cerebroventricular colchicine-induced memory impairment. In particular, it reduced AChE activity, improved OS (decrease of lipid peroxidation and nitrite levels, GSH restoration), and restored mitochondrial respiratory enzyme complex (I–IV) activities [288].

## 6. CA and BCP: Therapeutic Potential in Eye-Related Diseases

Glaucoma is a spectrum of progressive optic neuropathies causing vision loss through retinal ganglion cell (RGC) death and optic nerve degeneration. It shares common pathophysiologic mechanisms (e.g., alterations in brain fluid balance, the diseases’ spread via trans-synaptic degeneration, and apoptosis) with AD and PD. Accordingly, it can be considered a full-fledged neurodegenerative disease associated with aging [289,290,291]. Given its special connection via the optic nerve, the eye provides a unique gateway to the brain. Thus, glaucomatous damage not only affects the optic nerve but also different regions of the central nervous system, including the lateral geniculate nucleus, the intracranial optic nerves, and the visual cortex [292,293]. On the other hand, AD and glaucoma may be thought of as two clinical manifestations of a single NDD of the CNS [294]. In recent studies, patients with AD, PD, and Huntington’s disease (HD) showed microvascular and structural alterations of the retinal nerve fibre layer [295,296]. Marked upregulation of neuroinflammation can also occur in the retina, as a projection of the CNS. In this sense, glial cells are clearly involved in RGC health and disease, contributing to the release of inflammatory mediators and retinal degeneration in the pathophysiology of glaucoma. As reviewed elsewhere, mitochondria are central for RGC health, since their dysfunction is crucially involved in apoptosis of RGCs and axon degeneration in glaucoma [297,298]. Age-related macular degeneration (AMD; two forms, the *advanced nonexudative* or “*dry*” and the *neovascular* or “*wet*”) and retinitis pigmentosa (RP) also exhibit retinal degeneration as general pathophysiological aspects, being characterized by the loss of retinal pigment epithelium (RPE) cells and photoreceptors, as well as the development of protein deposits known as *drusen*. The major risk factors for the development of AMD and RP are genetics, aging, and environmental insults (e.g., cigarette smoking), inducing high levels of oxidative and nitrosative redox stress [299]. Oxidative injury can speed up RPE and photoreceptor cell death, as well as exacerbate angiogenesis and inflammation levels. On the other hand, it is becoming increasingly evident that microglia-mediated neuroinflammation is a key determinant of neuronal damage in AMD [300].

Accordingly, BCP and CA treatments may attenuate the level of OS/inflammation and prevent retinal degeneration.

In an in vitro study, CA treatments (10 µM) for 21 h protected retina-derived cell lines, ARPE-19 (RPEs) and 661W photoreceptors, against H_2_O_2_-mediated oxidative damage and cell death. In this regard, cytoprotection is afforded via Nrf2-mediated induction of phase 2 antioxidant enzymes, resulting in the reduction of hyperoxidized peroxiredoxin 2 (Prx2) formation [301]. Additionally, in a rat model of light-induced retinal degeneration (LIRD), CA protects retinas after light damage by preventing loss of photoreceptor functions [301]. Similarly, CA provides morphological and functional preservation of rod photoreceptors in an RP mouse model (rd10). In particular, CA-mediated neuroprotection is exerted through inhibition of retinal cell death, OS, inflammation, and reticulum stress [302]. Additionally, CA treatment triggered a marked increase of retinal Nrf2 in a light-induced damage mouse model and was effective in slowing (transcription factor myocyte enhancer factor 2d) Mef2d-haploinsufficient photoreceptor cell death [303]. In a rat model of acute or chronic photooxidative retinal damage, long-term dietary administration of an Age-Related Eye Disease Study (AREDS) antioxidant formulation supplemented with CA enhanced photoreceptor cell survival [304]. Moreover, CA treatment attenuated RPE cell death from acrylamide (ACR)-induced oxidative damage by enhancing SOD and catalase activities, the level of GSH, and the expression of Nrf2-induced antioxidant genes [305]. Therefore, when zebrafish embryos were exposed to ACR-induced retinal oxidative injury, the simultaneous presence of CA significantly reverted the toxic effects, including morphological abnormalities, loss of rod and cone photoreceptor cells, overgeneration of ROS, and decreased antioxidant capacities [305]. Another study revealed that a supplement mixture containing CA modulated human monocyte-derived macrophage (hMDM) functions from patients with AMD in in vitro and in vivo models that recapitulated features of AMD [306]. In particular, such formula reduced the angiogenic effects and modulated the neurotoxic actions of the treated M2a-polarized hMDMs ex vivo in choroidal sprouting and retinal explant assays. Moreover, the mixture also decreased the oxidative retinal injury in the photic injury model.

In rats with acute ocular hypertension, CA also promoted protective effects on the oxidative damage of RGCs [307]. Finally, treatment of chlorpyrifos-intoxicated mice with CA (at 30 and 60 mg/kg/day for 14 days) markedly reduced the presence of MDA and NO in ocular and brain tissues and increased the non-enzymatic (GSH) and enzymatic (GPx, SOD, and CAT) antioxidant defences [308]. In this regard, CA also decreased the serum levels of proinflammatory cytokines (IL-1β, IL-6, and TNF-α).

For BCP, no relevant literature data are available to date. However, some information on *Peperomia pellucida* extract may suggest the potential role of BCP in managing eye diseases. *Peperomia pellucida* (L.) Kunth is traditionally used as a medicinal plant by rural communities in tropical and subtropical regions, especially to manage inflammatory diseases including glaucoma, cataracts, and diabetic retinopathy [309]. Many bioactive compounds are found in *P. pellucida*, including BCP. Pharmacological mechanisms of action of *P. pellucida* include anti-inflammatory, antioxidant, antihypertensive, antihyperglycemic, and anti-angiogenic activities. Therefore, it will be interesting to investigate the bioactivity of BCP/CA, individually or in combination with current pharmaceuticals used for the treatments of eye-related diseases. Figure 5 shows the therapeutic potential of BCP and CA in the ocular system.

## 7. Conclusions and Future Perspectives

Nowadays, CA and BCP have caught a great deal of attention due to their ability to reduce OS-induced free radicals via an antioxidant effect and in managing inflammation through the control of several pathological pathways, as described in different experimental models. Moreover, when compared to other natural compounds, CA acts at a very low concentration range. Therefore, both compounds show high BBB permeability and afford a safe approach to protect against the neuronal damage in neurodegenerative diseases via the suppression of microglia-mediated neuroinflammation, as well as in promoting mitochondrial protection. They are effective in modulating mitochondrial quality control, especially mitophagy and mitochondrial biogenesis, and the mitochondria-dependent pathway involving antioxidant, anti-apoptotic, and bioenergetic effects (Figure 3 and Figure 4). Therefore, although further experimental studies and clinical trials are needed to better evaluate the potential neuroprotective efficacy, BCP and CA may represent promising candidates to prevent or delay the onset of neurological disorders. In this regard, given the lack of toxicity and easy availability from natural resources, it will be crucial to explore the chance of their direct employment (individually and/or in combination) as supplements in clinical practice. Successes in clinical trials would allow for the evaluation and elaboration of new nutritional intervention programs aimed at reducing the oxidative neuroinflammatory damage and decreasing/attenuating the progression of neurodegenerative diseases. Finally, considering the efficacy of various antioxidant/anti-inflammatory agents in preventing ocular disorders [310], the topical application of BCP and/or CA may be an alternative treatment option for promoting eye health.

## Figures and Tables

**Figure 1 antioxidants-11-01199-f001:**
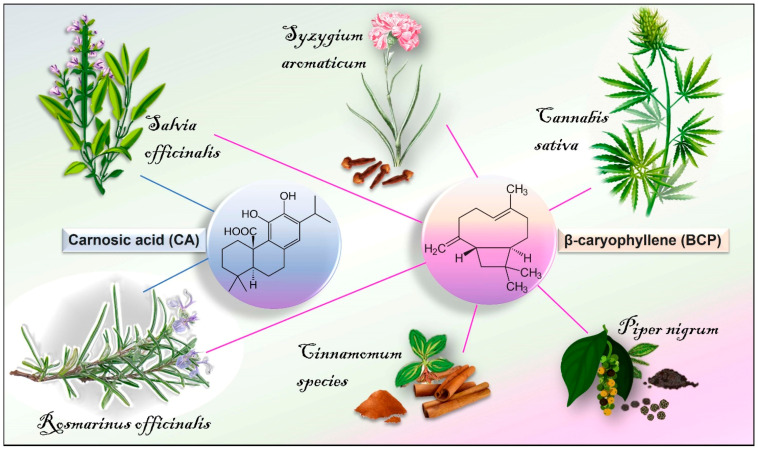
Chemical structures and main natural sources of β-caryophillene and carnosic acid.

**Figure 2 antioxidants-11-01199-f002:**
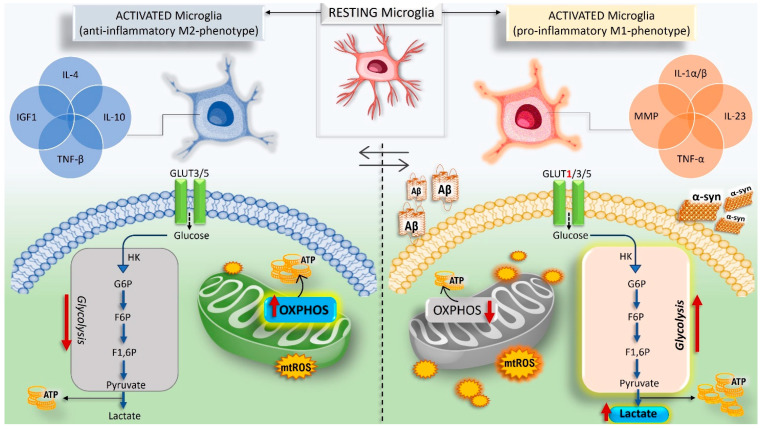
Schematic representation of microglial polarization, metabolic reprogramming, and immune responses under physiological and neuropathological conditions. In response to different environmental and cellular stresses, microglia promptly activate pro- or anti-inflammatory states to preserve tissue homeostasis. In addition to changes in morphology, phagocytosis capacity, and secretion of cytokines, activation of microglia results in changes in cellular energy demand. Under neuropathological conditions (e.g., Parkinson’s disease (PD) and Alzheimer’s disease (AD)), pro-inflammatory microglial activation (M1-like) requires metabolic shifts towards glucose uptake and glycolysis, with the consequent expression of glucose transporter 1 (GLUT1), as well as the production of lactate and reactive species. Conversely, under physiological and anti-inflammatory conditions, microglia (M2-like) mostly rely on oxidative phosphorylation.

**Figure 3 antioxidants-11-01199-f003:**
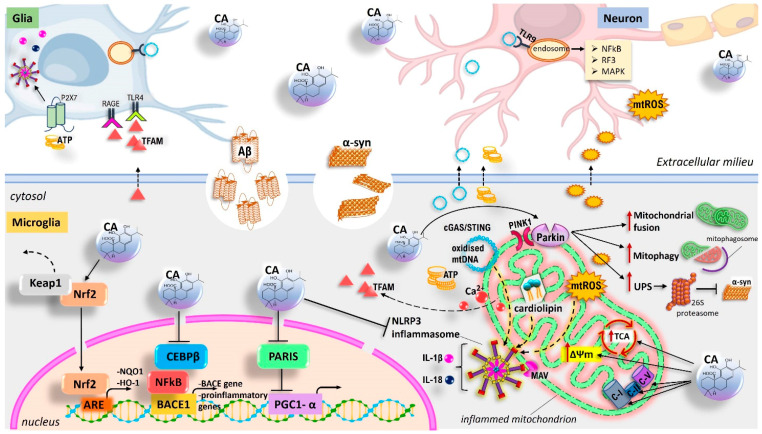
Current model of multi-target protective effects of carnosic acid (CA) at the interface of mitochondrial dysfunction, neuroinflammation, and neurodegeneration. Upon stressful conditions, including oxidative injury, accumulation of altered/misfolded proteins, and inflammatory stimuli, mitochondria are impaired with ensuing release of mitochondrial-derived DAMPs (mtDAMPs) such as ROS, Ca^2+^, oxidized mtDNA, cardiolipin, ATP, and transcription factor A mitochondria (TFAM). In surrounding neurons and glial cells, mtDAMPs bind to TLR9, purinergic P2X7 receptor, RAGE, and TLR, promoting further mitochondrial damage and triggering pro-inflammatory, pro-apoptotic cascades that contribute to exacerbation of proinflammatory microglial activation, with the consequent degeneration of dopaminergic neurons. CA provides neuroprotective effects, decreasing oxidative stress by upregulating antioxidant defence, reducing the expression of proinflammatory cytokines via inhibiting the NFκB pathway and/or NLRP3 inflammasome activation, and preserving mitochondrial function. Therefore, Keap1 degradation and Nrf2/HO-1/NQO1 signalling pathway activation, upregulation of the PINK1/parkin pathway, and modulation of the PARIS/PGC-1α axis contribute to counteracting oxidative stress and energy imbalance, as well as mitochondrial disruption by promoting mitophagy, mitobiogenesis, mitochondrial fusion, and UPS machinery. Additionally, CA decreases the CEBPβ–NFκB interaction and reduces Aβ aggregation/deposition by inactivating the amyloidogenic proteolytic pathway of amyloid precursor protein (APP).

**Figure 4 antioxidants-11-01199-f004:**
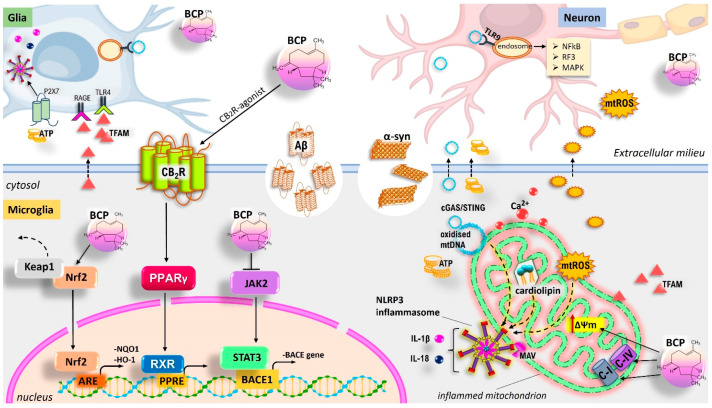
Current model of multi-target protective effects of β-caryophillene (BCP) at the intersection of mitochondrial dysfunction, neuroinflammation, and neurodegeneration. Upon stressful conditions, including oxidative injury, accumulation of altered/misfolded proteins, and inflammatory stimuli, mitochondria are impaired with the ensuing release of mtDAMPs such as ROS, Ca^2+^, oxidized mtDNA, cardiolipin, ATP, and TFAM. In surrounding neurons and glial cells, mtDAMPs bind to TLR9, the purinergic P2X7 receptor, RAGE, and TLR, promoting further mitochondrial damage and triggering pro-inflammatory, pro-apoptotic cascades that contribute to exacerbation of proinflammatory microglial activation, with the consequent degeneration of dopaminergic neurons. BCP provides neuroprotective effects, decreasing oxidative stress by upregulating antioxidant defence, reducing the expression of proinflammatory cytokines by inhibiting the NFκB pathway, and preserving mitochondrial function upregulating respiratory enzyme complex (I–IV) activities. Therefore, Keap1 degradation and Nrf2/HO-1/NQO1 signalling pathway activation, upregulation of the CB2R/PPARγ axis, and inhibition of the “JAK2-STAT3-BACE1” signalling pathway suppress TLR4, NO, PGE2, iNOS, and COX-2 expression, the secretion of pro-inflammatory cytokines, as well as Aβ oligomerization and deposition.

**Figure 5 antioxidants-11-01199-f005:**
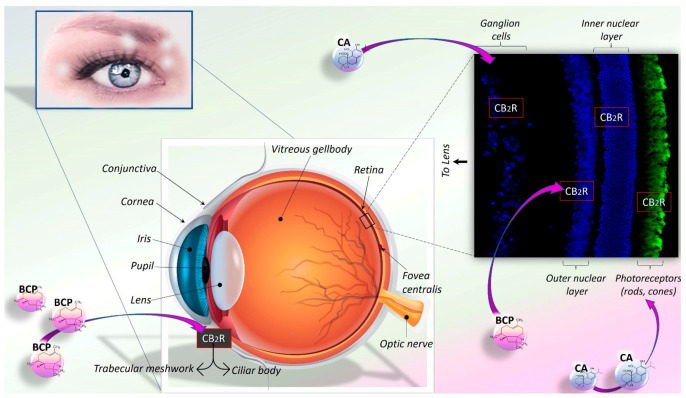
Potential hypotensive and neuroprotective actions of carnosic acid and β-caryophillene in ocular tissues. Considering the localisation and the functional involvement of CB_2_R in the trabecular meshwork, ciliary body, and retina, BCP-mediated beneficial effects could be hypothesized. Potential neuroprotective efficacy of CA has also been demonstrated in different ocular tissues, including retinal ganglion cells and photoreceptors.

## Data Availability

Data is contained within the article.

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
