# Peer review of "Multi-Target Effects of ß-Caryophyllene and Carnosic Acid at the Crossroads of Mitochondrial Dysfunction and Neurodegeneration: From Oxidative Stress to Microglia-Mediated Neuroinflammation"

_antioxidants, 2022, doi:10.3390/antiox11061199_

Round 1

Reviewer 1 Report

In this paper, authors proposed how  ß-caryophyllene and carnosic acid exert neuroprotective and anti-inflammatory effects in chronic neurodegenerative disorders such as AD, PD, and retinal degeneration.

The association of mitochondrial dysfunction and NLRP3 inflammasome activation in microglia-mediated neuromodulation and neuroinflammation was also discussed.

Authors suggest that both compounds could be a promising strategy in the management of neurodegenerative diseases aimed at maintaining mitochondrial homeostasis and ameliorating glial-mediated neuroinflammation.

Considering the depth quality and completeness of the paper, it is considered that this review paper deserves publication in the antioxidants journal.

Author Response

We thank the reviewer for his/her useful comments. Please find below a point-to-point answer to their questions. We have also improved the review with the Graphical Abstract, and the Figure 5.

All coauthors have agreed to the revisions.

REVIEWER#1

Response: We are pleased to thank the reviewer for his/her comments

Reviewer 2 Report

This is a well written review on ß-Caryophyllene (BCP) and Carnosic Acid (CA), which have anti-inflammatory, and antioxidant activities, as well as neuroprotective and mitoprotective effects in different in vitro and in vivo models.

BCP and CA are very well described as well as their biological activities. Lot of appropriated references are cited.

The quality of figures are also of high qualities.

However, the legends of Figures 2, 3 and 4 does permit to understand the corresponding figures. In the corresponding legends, I suggest to the authors to briefly explain the figures. The legends must be brief and informative. They must allow to easily understand each figure without a need to read the manuscript (10-20 sentences per figure legend are required).

Author Response

We thank the reviewer for his/her useful comments. Please find below a point-to-point answer to their questions. We have also improved the review with the Graphical Abstract, and the Figure 5.

All coauthors have agreed to the revisions.

REVIEWER#2

……….However, the legends of Figures 2, 3 and 4 does permit to understand the corresponding figures. In the corresponding legends, I suggest to the authors to briefly explain the figures. The legends must be brief and informative. They must allow to easily understand each figure without a need to read the manuscript (10-20 sentences per figure legend are required).

Response: We are pleased to thank the reviewer for his/her comments.

Regarding the figure legends we would like to explain that in the first stage of the submission a system error occurred: in the conversion of the file, the text of the figure legends (figures 2, 3, and 4) was hived off from their title.  However, we have promptly notified the associate editor of the error. We apologize for the misunderstanding. In the revised manuscript you can find the complete figure legends.

Reviewer 3 Report

This is an extensive review by Iorio et al, regarding the effects of ß-Caryophyllene and Carnosic Acid on mitochondrial dysfunction in neurodegeneration. The review covers descriptions of the potential medicinal therapeutics to treat neuroinflammation and degeneration via suppressing oxidative stress, the structural details of the agents, and the molecular mechanisms associated with therapy. Figures are detailed and self-explanatory and make it easier for the readers to understand the topic. Except that the title can be a little shorter, less wordy, and simple, I do not have any additional comments to improve the submission.

Author Response

We thank the reviewer for his/her useful comments. Please find below a point-to-point answer to their questions. We have also improved the review with the Graphical Abstract, and the Figure 5.

All coauthors have agreed to the revisions.

REVIEWER#3

This is an extensive review by Iorio et al, regarding the effects of ß-Caryophyllene and Carnosic Acid on mitochondrial dysfunction in neurodegeneration. The review covers descriptions of the potential medicinal therapeutics to treat neuroinflammation and degeneration via suppressing oxidative stress, the structural details of the agents, and the molecular mechanisms associated with therapy. Figures are detailed and self-explanatory and make it easier for the readers to understand the topic. Except that the title can be a little shorter, less wordy, and simple, I do not have any additional comments to improve the submission.

Response: We are pleased to thank the reviewer for his/her suggestions.

With regard to the title, we agree with the reviewer’s comment. However, we would like to maintain our proposed title. In our opinion, given the extensive scope of the review (as the reviewer has already rightly said), it would include all the topics covered. Therefore, also in light of added graphical abstract and figure 5, we kindly ask you to accept the title in its current form.